# Evolution of the locomotor skeleton in *Anolis* lizards reflects the interplay between ecological opportunity and phylogenetic inertia

Nathalie Feiner [1✉], Illiam S. C. Jackson [1,2], Edward L. Stanley [3] & Tobias Uller[1]

*Anolis* lizards originated in continental America but have colonized the Greater Antillean islands and recolonized the mainland, resulting in three major groups (Primary and Secondary Mainland and Greater Antillean). The adaptive radiation in the Greater Antilles has famously resulted in the repeated evolution of ecomorphs. Yet, it remains poorly understood to what extent this island radiation differs from diversification on the mainland. Here, we demonstrate that the evolutionary modularity between girdles and limbs is fundamentally different in the Greater Antillean and Primary Mainland *Anolis*. This is consistent with ecological opportunities on islands driving the adaptive radiation along distinct evolutionary trajectories. However, Greater Antillean *Anolis* share evolutionary modularity with the group that recolonized the mainland, demonstrating a persistent phylogenetic inertia. A comparison of these two groups support an increased morphological diversity and faster and more variable evolutionary rates on islands. These macroevolutionary trends of the locomotor skeleton in *Anolis* illustrate that ecological opportunities on islands can have lasting effects on morphological diversification.

[1] Department of Biology, Lund University, Lund, Sweden. [2] College of Natural Sciences, University of Texas at Austin, Austin, TX, USA. [3] Department of Natural History, Florida Museum of Natural History, University of Florida, Gainesville, FL, USA. ✉email: nathalie.feiner@biol.lu.se

Lineages that colonize islands often rapidly diversify along distinct evolutionary trajectories, as famously demonstrated by the Darwin Finches[1] and Hawaiian silverswords[2]. Important reasons for this include that islands can harbor a range of ecological settings, few competing species, and low predation[3,4]. In comparison, lineages that establish on the mainland typically face more limited opportunities to diversify since ecological niches tend to be already occupied by similar organisms, and competition and predation may be severe.

Ecological opportunities do not exist in isolation, however, and they will be realized only insofar as there are phenotypes that can exploit them[5]. In the short term, the capacity to adapt is limited by the phenotypes that can be generated from standing genetic variation[6,7]. In the longer term, adaptive diversification depends on the capacity of development to generate phenotypes that can interact with the environment in novel ways[8–10]. Differences in development, physiology, and behavior can lead to persistent differences between clades in the extent to which parts of the organism evolve together (i.e., evolutionary modularity and integration[11], reviewed in refs. [12,13]). Yet, the relationship between adaptive diversification and evolutionary modularity remains poorly understood[11,14–18]. On the one hand, diversification into novel ecological opportunities may require changes in variational properties, in particular less constrained covariation between parts[19–22]. For example, primates specialized in vertical clinging and leaping (e.g., lemurs and tarsiers) have weaker phenotypic integration between fore- and hindlimbs compared to other quadruped primates[23]. In contrast, the extraordinary diversification of beak and skull shape in Hawaiian honeycreepers and Darwin finches proceeded along the same evolutionary covariance as other birds, illustrating that the existing developmental and functional integration of the avian head is fully capable of generating extreme morphologies[24].

The biogeographic history of the species-rich Anolis lizards offers an outstanding opportunity to explore how ecological opportunity and developmental bias shape adaptive diversification. Early in the history of the Anolis genus, original mainland forms (hereafter Primary Mainland) from continental America colonized Greater Antillean islands where they diversified into more than 100 species (Fig. 1a). Subsequently, Anolis lizards most closely related to extant Jamaican species dispersed back to Central and South America (hereafter Secondary Mainland) and gave rise to over 100 extant species[25–27]. The numerous small islands of the Lesser Antilles that typically contain only one or two species per island were colonized in two waves, one early wave from the Primary Mainland clade, and one later wave from the Greater Antilles[28] (Fig. 1a).

Greater Antillean Anolis have produced neither more species nor an overall higher diversity in gross morphology than mainland anoles[29–31]. However, the Greater Antillean and mainland Anolis appear to differ in the functional relationship between morphology (e.g., relative limb length) and aspects of the lizards' ecology (e.g., perch diameter)[32–34]. In particular, the Greater Antillean islands are characterized by the presence of up to six ecomorphs, each adapted to a certain microhabitat that imposes distinct functional demands on locomotion[35–37].

In this work, we test if the adaptive radiation on the Greater Antilles was accompanied by faster or more variable evolutionary rates and increased morphological disparity of the locomotor skeleton than on the mainland. Further, we test if island and mainland diversifications have proceeded along similar and deeply conserved patterns of covariation in the locomotor skeleton, or if the adaptive radiation on the Greater Antilles was accompanied by changes in evolutionary modularity and integration. The results reveal that the evolutionary modularity of limbs and girdles indeed differs fundamentally between Greater Antillean Anolis and Primary Mainland Anolis. However, the evolutionary modularity of Greater Antillean Anolis was shared with the group that recolonized the mainland, a pattern accompanied by higher morphological diversity and faster and more variable evolutionary rates on islands. These macroevolutionary trends illustrate how morphological diversification is shaped by the interplay between ecological opportunity and phylogenetic inertia.

## Results

A phenotyping of 704 individuals from 271 species (including four closely related non-Anolis species) allowed us to unravel patterns of evolutionary diversification in the locomotor skeleton of Anolis lizards. Specimens from museum collections were scanned using microcomputed tomography (micro-CT). We used 3D geometric morphometrics to capture variation in the shape of the pectoral and pelvic girdles with 18 landmarks each (Supplementary Table 1)[38,39], as well as univariate length measurements of 15 limb elements that capture morphological variation in the fore- and hindlimbs relative to body size (Supplementary Fig. 1). We supplemented these four blocks (pelvic girdle, pectoral girdle, forelimb, and hindlimb) with centroid size as a proxy for body size, resulting in a dataset comprising 124 features or traits. To allow an inclusive analysis of the entire locomotor skeleton, we standardized (z-transformed) the 124 traits to account for the fact that they are not on a commensurate scale[40]. This procedure removes certain properties (e.g., the original trait variances) from the dataset[40,41], but the transformation allows us to infer morphological differences among Anolis groups across the entire locomotor skeleton, which is our primary focus. Analyses that would be compromised if performed on standard normal deviates (e.g., disparity analysis)[40] were performed on girdles and limbs separately to allow retention of original trait variances[42].

### Greater Antillean Anolis show greater morphological disparity than the Secondary Mainland clade.

Anolis species belonging to the Greater Antilles, Lesser Antilles, and the Primary and Secondary Mainland groups occupy largely overlapping regions in morphospace, although slight differences exist (Fig. 1b, c, and Supplementary Fig. 2). Variation in PC1 and PC2 is associated with the shape of the pelvic and pectoral girdle, respectively, whereas variation in PC3 is also associated with variation in the relative lengths of limb bones (Supplementary Table 2). Body size does not load strongly on any of the first three PCs.

The evolution of the locomotor skeleton is characterized by a significant phylogenetic signal ($K_{mult\_total} = 0.571$; $K_{mult\_girdles} = 0.586$; $K_{mult\_limbs} = 0.945$; all $P < 0.001$), which justifies taking a formal comparative phylogenetic approach (see "Methods"). Using this framework, we find that species of the Greater Antilles occupy a larger volume in morphospace (Fig. 1b, c and Supplementary Fig. 2) and show higher morphological disparity (Procrustes variance ($PV$)$_{girdles} = 0.024$; $PV_{limbs} = 0.051$) than species inhabiting the Lesser Antilles ($PV_{girdles} = 0.015$, $P_{girdles\_GAvsLA} = 0.006$; $PV_{limbs} = 0.011$, $P_{limbs\_GAvsLA} = 0.002$) and species of the Secondary Mainland clade ($PV_{girdles} = 0.017$, $P_{girdles\_GAvsMLsec} < 0.001$; $PV_{limbs} = 0.038$, $P_{limbs\_GAvsMLsec} = 0.036$). In contrast, the morphological disparity of the Primary Mainland clade is on par with the high levels attained by the Greater Antillean group ($PV_{girdles} = 0.022$, $P_{girdles\_GAvsMLpri} = 0.460$; $PV_{limbs} = 0.047$, $P_{limbs\_GAvsMLpri} = 0.707$; Supplementary Tables 3 and 4). Thus, the colonization of the Greater Antillean islands was not accompanied by a more extensive exploration of morphospace than on the mainland, but the morphological diversification of the locomotor skeleton was substantially reduced in the clade that recolonized the mainland (and in the groups that colonized the Lesser Antilles).

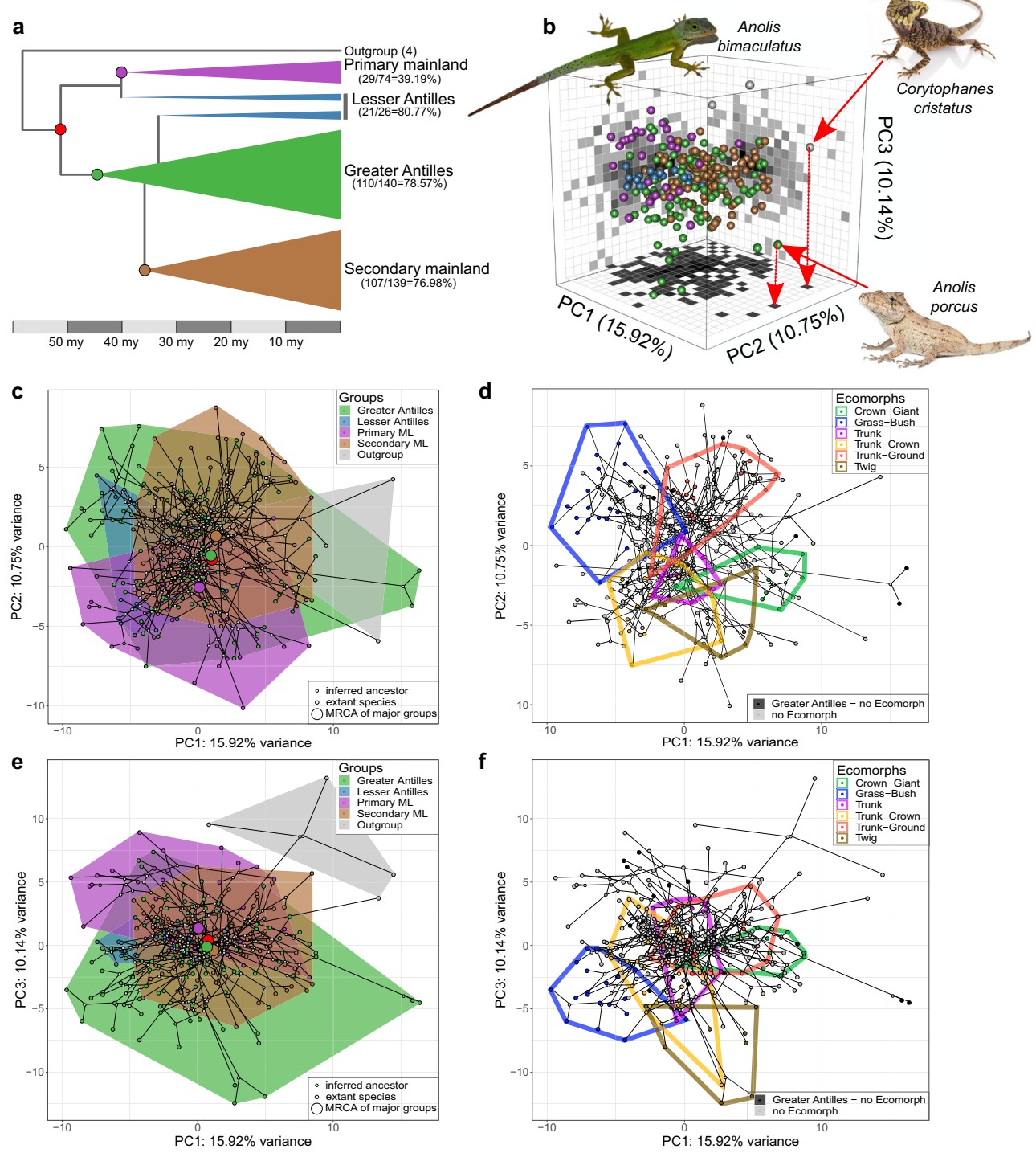

One possible explanation for the difference between the Greater Antillean and Secondary Mainland groups is that only the former has evolved distinct adaptations to different microhabitats (i.e., ecomorphs)[29,32]. Different microhabitats impose different functional demands on locomotion, and the ecomorph classification indeed explained 9% of morphological variation in the locomotor skeleton in the Greater Antillean *Anolis* (Supplementary Table 5). The exaggerated morphology of species that belong to the grass-bush, crown-giant (both mainly PC1), and twig (mainly PC3) ecomorphs appears to be absent from both the Primary and Secondary Mainland clades (Fig. 1d, f, and Supplementary Fig. 2). Furthermore, the Greater Antillean sister

species *Anolis porcus* and *Anolis chamaeleonides* show an extreme morphology compared to all other extant *Anolis* (Fig. 1b and Supplementary Fig. 2), which is closer to the distant relatives that shared an ancestor with *Anolis* more than 60 million years ago (Fig. 1b)[43]. While the Lesser Antillean group does not appear to possess any unique morphologies, the Primary Mainland *Anolis* exhibit morphologies of the pectoral girdle (PC2) and limbs (PC3) that are absent in all the other groups (Fig. 1c, e, and Supplementary Fig. 2).

Based on this detailed description of the locomotor skeleton of *Anolis* lizards, we sought to establish whether the group differences in morphological disparity and their overlap in

**Fig. 1 Morphospace of the locomotor skeleton of *Anolis* lizards. a** Phylogenetic relationship between major groups of *Anolis* lizards. The number of species per group included in this study, the total number of species, and the resulting percentages, are given in brackets and are proportional to the height of the triangles in the phylogram. The deepest split within *Anolis* marks the dispersal of mainland forms to the Greater Antillean islands. One lineage derived from the Greater Antillean lineage has recolonized the mainland (Secondary Mainland clade). The Lesser Antilles has been colonized from two different sources, once from the Primary Mainland clade and once from the Greater Antillean lineage. The timeline indicates the age of the major divergences as reported by Poe et al.[26]. **b** The first three principal components of the locomotor skeleton of all 271 species included in this study in a 3D morphospace. The two sister species, *A. porcus* and *A. chamaeleonides*, have undergone an extreme shift in morphospace along the first PC, approaching the position of distantly related genera. Pictures show an *A. bimaculatus* representing the second most "average" *Anolis* species (i.e., closest to the centroid described by PC1–3 of all *Anolis* species), a *Corytophanes cristatus* representing distant relatives to *Anolis*, and an *A. porcus* representing the two species with the most extreme shifts in morphospace along PC1. **c** The first and second PCs visualizing morphospace occupancy of *Anolis* species color-coded by group. The 110 species of the Greater Antilles show a higher morphological disparity compared to the 107 species of the Secondary Mainland clade and to the 21 species of the Lesser Antilles (see also and Supplementary Tables 3 and 4). Large circles mark the inferred position of the most recent common ancestor (MRCA) of the three major groups and of all *Anolis* species (in red, see also panel **a**). **d** The first and second PCs visualizing morphospace occupancy of *Anolis* species color-coded by ecomorph. **e**, **f** The first and third PCs color-coded by biogeographic group and ecomorph, respectively. See Supplementary Fig. 2 for a more detailed exploration of the PC analyses. Picture credit: *C. cristatus*: picture sourced from "Nature Picture Library"; *A. bimaculatus*: picture sourced from https://commons.wikimedia.org/ under the licence CC BY 2.0 (https://creativecommons.org/licenses/by/2.0/deed.en; creator: Clinton and Charles Robertson; without modifications of the original image); *A. porcus*: picture sourced from "Alamy Limited". Abbreviation: ML mainland.

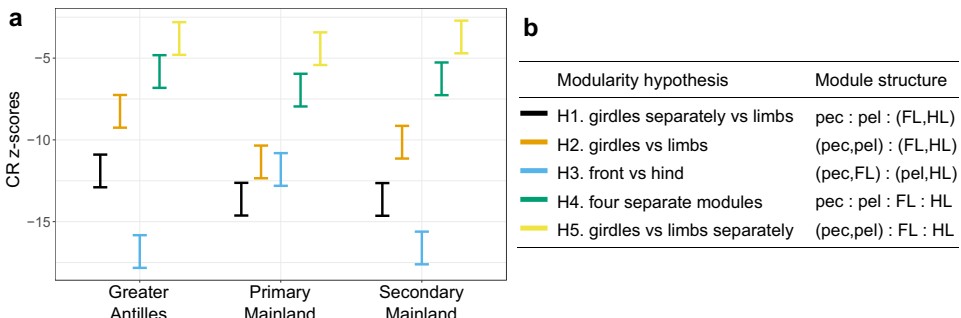

**Fig. 2 Evolutionary modularity is shared between the Greater Antillean and Secondary Mainland groups while the Primary Mainland is distinct. a** CR *z*-scores (effect sizes) of the five alternative modularity hypotheses, evaluated separately for the three major groups. CR *z*-scores that are more negative signify a stronger modularity (i.e., stronger covariation within modules relative to the covariation between modules). CR *z*-scores are derived from a permutation approach (here, 1000 iterations) to derive an empirical null distribution against which the observed CR value of each configuration is compared (see "Methods" for more details). **b** Configurations of the five alternative modularity hypotheses. The modularity hypothesis H1, limbs forming a single evolutionary module, had the highest support (i.e., lowest CR *z*-score) in the Primary Mainland clade. In contrast, the modularity hypothesis H3 with pectoral girdle and forelimb forming a front module and pelvic girdle and hindlimb forming a hind module, had a significantly higher support (i.e., lower CR *z*-score) than the other hypotheses in the Greater Antillean group and the Secondary Mainland clade. For statistical support of alternative models, see Supplementary Tables 6–8. Abbreviations: FL forelimb, HL hindlimb, pec pectoral girdle, pel pelvic girdle.

morphospace are accompanied by (a) differences in the evolutionary modularity and integration, and (b) changes in evolutionary rates of the locomotor skeleton. Since the Lesser Antillean species are divided into two small and distantly related clades, each with relatively few species (Fig. 1a), we excluded them from the following comparisons and instead focused on the three major groups.

**Group differences in modularity structure.** We considered five modularity hypotheses for the evolutionary covariation between the four blocks of the locomotor skeleton (i.e., the pelvic and pectoral girdles and the front- and hindlimbs), and assessed their support in each of the three major groups. First, since the elements of the fore- and hindlimbs share the same deeply conserved developmental genetic architecture[44–46], they are expected to coevolve more tightly than other parts of the locomotor skeleton and form one single evolutionary module (with the pelvis and pectoral forming two separate modules (H1) or a single module (H2), Fig. 2). These two hypotheses describe the perhaps most intuitive covariance structure. However, since limbs and their respective girdles show a strong functional dependency, evolutionary diversification could proceed via a stronger

correlation between limbs and their respective girdles than between the limbs themselves, resulting in two evolutionary modules (H3 in Fig. 2). Alternatively, the large variation in the locomotor skeleton within squamate reptiles could imply that the pectoral and pelvic girdles and front and hindlimbs are all free to evolve independently of each other (i.e., four modules; H4 in Fig. 2). Finally, girdles could form a single module, while fore- and hindlimbs evolve as two independent modules (H5 in Fig. 2). To sort between these hypotheses, we performed tests of modularity using the covariance ratio (CR) approach[47,48] that assesses the strength of covariation *between* modules relative to *within* modules and generates an effect size estimate (CR *z*-score) derived from a permutation procedure[49] (see "Methods"). By using a subsampling procedure, we established that uneven species coverage does not bias these estimates (Supplementary Fig. 3a, b), and therefore present the results including all available species for each of the three groups.

The Primary Mainland clade follows the prediction derived from the shared genetic architecture of limbs, with the best-supported modularity structure being that of limbs forming one single module, and pectoral and pelvic girdles two additional modules (H1 in Fig. 2 and Supplementary Table 6). In contrast,

the Greater Antillean and the Secondary Mainland groups have evolved according to a fundamentally different modularity structure that separates a front (pectoral girdle and forelimbs) and a hind (pelvic girdle and hindlimbs) module (H3 in Fig. 2a and Supplementary Tables 7 and 8). This result persisted if the 10% of species with the most extreme morphologies of the Greater Antillean group were excluded, which demonstrated that the modularity structure is not driven by exaggerated morphologies (Supplementary Table 9).

Given these differences in modularity structure between the major groups of *Anolis*, we proceeded to compare the strength of pairwise integration between all four blocks in each of the major groups. We used phylogenetic partial least-square (PLS) analyses coupled with effect-size comparisons (PLS z-scores) following Adams and Collyer[47]. Since incomplete sampling can bias the estimated strength of integration, with less dense species sampling underestimating effect sizes (Supplementary Fig. 3c), we equalized the proportion of species sampled between the three major groups by rarefying the proportion of species sampled[50] for each major group to 39% (see Fig. 1a) over 1000 iterations (see Methods).

In accordance with the distinct modularity structure of the locomotor skeleton in the Primary Mainland clade, the integration between limbs and their respective girdles is generally lower in this clade than in the Greater Antillean *Anolis* (statistically significant in 49.3% [front module] and 18.6% [hind module] of subsampled datasets; Supplementary Fig. 4c, d). For the Secondary Mainland, this was only evident for the front module (significant in 41.3% of subsampled datasets; Supplementary Fig. 4c). There was no systematic difference in the strength of integration among the four blocks between the Greater Antillean group and the Secondary Mainland clade (Supplementary Fig. 4).

**Greater Antillean *Anolis* evolved at a more variable rate**. Given that the overall modularity structure was conserved between the Greater Antillean and Secondary Mainland groups, we sought to test if Greater Antillean *Anolis* have attained a higher morphological disparity by evolving at consistently higher or more variable evolutionary rates.

Body size has evolved three times faster in Greater Antillean *Anolis* ($\sigma^2 = 715.96$) than in *Anolis* belonging to the Secondary Mainland clade ($\sigma^2 = 238.97$, Supplementary Table 10). The net evolutionary rate of body size in the Primary Mainland *Anolis* ($\sigma^2 = 603.86$) approaches the high net rates of the Greater Antillean species, suggesting a slowing down of body-size evolution in the Secondary Mainland *Anolis* rather than an acceleration in Greater Antillean *Anolis*.

To model the temporal evolutionary dynamics in the locomotor skeleton across the *Anolis* phylogeny, we used a Bayesian reversible-jump Markov chain Monte Carlo (rjMCMC) approach[51]. This method requires orthogonal variables, and we therefore used principle components (the number determined by the broken stick method[41]) of the full data set, as well as separately for limbs and girdles. While this does not retain original trait variances, and absolute rates therefore are not meaningful[40], it allows a comparison of relative rates and rate shifts between the three focal groups (see refs. 24,52,53 for similar approaches). We evaluated the overall levels of relative evolutionary rates as well as the number of rate shifts. Rate shifts are significant increases or decreases in rates, and we discriminate here between shifts associated with a single branch in the tree (branch shifts) and shifts that affect an entire clade (node shifts).

Across the *Anolis* phylogeny, there were more branch than node shifts (21 vs. 5, Fig. 3a). While branch shifts were evenly distributed across the tree, all but one node shift occurred in the

Greater Antillean group. Consistent with their extreme position in morphospace, the parental node of the Chamaeleonides group comprising *A. porcus* and *A. chamaeleonides* is one example of acceleration in evolutionary rate of the locomotor skeleton. Another example is the node from which the unusual twig ecomorphs *A. sheplani* and *A. placidus* evolved. The single node shift that occurred outside the Greater Antilles was detected at the base of the Primary Mainland clade (Supplementary Fig. 5 and Supplementary Table 12). In addition to an accumulation of node shifts in the Greater Antillean group, we also found that the variance in evolutionary rates was significantly higher in the Greater Antillean group compared to the Secondary Mainland clade (robust Brown–Forsythe Levene-type test of homogeneity of variance followed by Tukey post hoc test, $P < 0.006$, Fig. 3b, Supplementary Fig. 5 and Supplementary Table 13). This effect was not driven by variation in body size and remained pronounced if limb length alone is considered, but failed to reach significance if only girdle shape was analyzed (Supplementary Fig. 5 and Supplementary Table 13). The higher variance in evolutionary rates in the Greater Antillean group compared to the Secondary Mainland clade was accompanied by an overall elevation of relative evolutionary rates in Greater Antillean *Anolis* (Supplementary Fig. 5, Supplementary Tables 12 and 13).

Taken together, these results suggest that the greater evolutionary disparity in Greater Antillean *Anolis* compared to the Secondary Mainland clade has accumulated through modestly elevated evolutionary rates, interspersed by occasional bursts that were most pronounced in the dimensions of the limb bones.

## Discussion

Establishing how traits vary together across a phylogeny can give important insights into the genetic, developmental, and functional interactions that generate phenotypic evolution[54]. Organisms are never designed from scratch, and adaptive change will necessarily be shaped by their developmental biology. Yet, as organisms adapt and diversify, their capacity for future evolution may change[13,55]. Our results demonstrate that the macroevolution of the locomotor skeleton of *Anolis* lizards reflects this interplay between ecological opportunity and phylogenetic inertia.

Roughly, 50 million years of *Anolis* evolution have produced a large number of species, but they all share distinct properties that make them recognizable as *Anolis*[29]. Yet, the Greater Antillean adaptive radiation appears to be unusual in that it is characterized by the repeated evolution of ecomorphs, a functional specialization to microhabitats that is less pronounced on the mainland[35–37] (as well as on the species-poor Lesser Antillean islands). Here, we show that this Greater Antillean adaptive radiation was characterized by a strong coevolution of limbs and girdles, but that it did not produce more diverse morphologies than the clade that remained on the mainland.

The developmental genetics of the locomotor skeleton suggests that the strongest covariance should be between elements of fore- and hindlimbs[44–46,56]. The evolutionary modularity of the Primary Mainland clade followed this expected pattern, suggesting that a close developmental genetic integration of limbs is ancestral also to *Anolis*. The shift in evolutionary modularity in Greater Antillean *Anolis* is further evidence that this adaptive radiation is special[29,32,36]. While the reasons for this shift cannot be confidently established from these data, one possible explanation is that Greater Antillean species are adapted to ecological conditions that are peculiar to islands. However, if ecological differences alone were responsible, the majority of lineages of the Greater Antillean group must have experienced more consistent

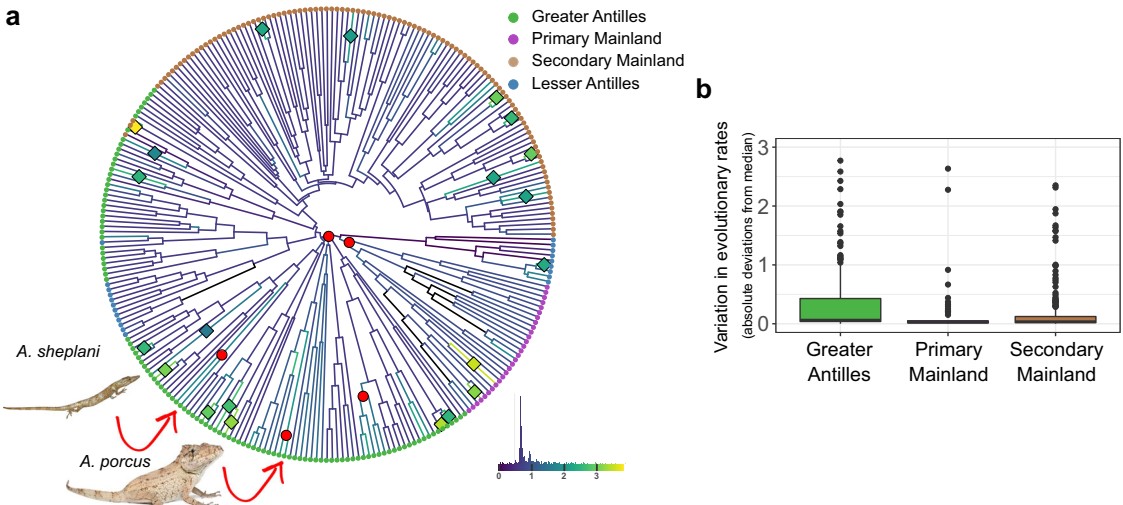

**Fig. 3 Evolutionary rates are more variable in Greater Antillean Anolis. a** Phylogenetic tree summarizing the results of the rjMCMC variable rate analysis estimating the propensity of edges and nodes to vary in evolutionary rates. Colored dots at the tips indicate to which biogeographic group a given species belongs. Diamonds in the phylogeny mark the position of branch shifts and red circles the position of node shifts that were identified in the majority of the posterior samples. Branch colors indicate relative evolutionary rates (log-transformed) and the histogram (bottom right) summarizes the frequency distribution of individual branches. Pictures show an *A. sheplani* (picture provided by Alejandro J. Sánchez) and an *A. porcus* (picture sourced from "Alamy Limited"). **b** Box plot summarizing the variance in evolutionary rates per branch of the full dataset as absolute deviations from the median for each major group. Colored boxes span the interquartile range with the median indicated by a horizontal line, whiskers represent 1.5 times the interquartile range, and outliers are represented by dots. Number of branches per group are $n = 229$ for Greater Antilles, $n = 58$ for Primary Mainland, and $n = 208$ for Secondary Mainland. Brown–Forsythe tests for equality of variance followed by Tukey post hoc tests (both two-sided) revealed that the Greater Antillean group showed a significantly higher variance than the Secondary Mainland clade (Supplementary Tables 12 and 13 give the full report of the statistical analyses).

correlational selection on limbs and girdles than did lineages of the Primary Mainland clade. Given that the structural habitats occupied by *Anolis* are rather similar on islands and the mainland[32], and that islands themselves harbor a variety of microhabitats[29], it is not obvious why this would be the case. It is therefore plausible that the distinct evolutionary modularity of the Greater Antillean group, relative to the Primary Mainland clade, in part reflects an ancient and persistent difference in how girdles and limbs develop and grow together. Such differences in how phenotypes are generated are known to influence how evolution proceeds[13], and will be reflected in covariation of traits across a phylogeny (i.e., evolutionary modularity and integration)[15,17,18,57]. For example, the peculiar reproductive biology of marsupials, like that of the kangaroo and its allies, is associated with weaker integration and increased modularity of fore- and hindlimbs both within and across species[58,59] (but see also ref. [60]).

A role for persistent bias in the generation of phenotypic variation is supported by the fact that the clade that recolonized the mainland exhibits an equally strong covariation between limbs and girdles as seen in the Greater Antillean radiation. Moreover, while the Secondary Mainland clade produced an equal number of species as the Greater Antillean group (and more species than the Primary Mainland clade), the morphologies of the locomotor skeleton are a subset of those that evolved on the Greater Antilles. In contrast, the overlap with morphologies of the Primary Mainland clade is modest.

These results are unexpected if consistent ecological differences (and therefore selective pressures) between island and mainland were the sole reason for the morphological differences between island and mainland *Anolis*. Yet, the results are intelligible if evolution on the Greater Antilles had persistent effects on the lizards' developmental and behavioral biology, thereby imposing a bias on their future evolution following re-colonization of the mainland. Theoretical models of evolvability have demonstrated

that strong selection for certain combinations of traits can promote the evolution of developmental interactions that make those traits vary together despite the genetic change being random[61–63] (reviewed in ref. [13]). Thus, adaptation in response to ecological opportunity following the colonization of the Greater Antilles could have resulted in a stronger developmental integration of limbs and their respective girdles. This, in turn, could have imposed a bias on future evolutionary change that persisted in lizards that recolonized the mainland, resulting in phylogenetic inertia[64]. Similar mainland and island comparisons of other defining features of the Greater Antillean adaptive radiation, such as the morphology of skulls, would be highly interesting.

The interplay between ecological and developmental causes of adaptation is not possible to disentangle on the basis of covariation across *Anolis* species alone, but requires comparison of phenotypic modularity and integration within and across species[9,11,16]. Studies of mammals, flies, and worms have indeed demonstrated that morphological diversification can proceed along "developmental lines of least resistance" that persist for many millions of years[65–68]. At the same time, selection can modify patterns of morphological integration on surprisingly short timescales[69,70]. For example, the within-species integration of the skull can vary between ecomorphs in Greater Antillean *Anolis*[64], but how these features of the skull coevolve across the phylogeny remains to be explored. With respect to the locomotor skeleton, it would be particularly interesting to compare patterns of morphological variation within species from each of the three major groups. While the literature emphasizes differences between island and mainland *Anolis*[30,32,34], our results predict that the two mainland clades consistently will differ in the covariation between limbs and girdles within species. Furthermore, if selection is able to modify skeletal development and growth[61,62,71], species with distinct morphologies, like the Chamaeleonides group of the Greater Antillean *Anolis*, may stand out in terms of morphological variability. Lizards from island and

mainland lineages may also interact differently with their environment, and more information on functional aspects of morphology, including how morphological differences impact on ecological performances, is sorely needed[32–34,72]. However, plastic responses to different microhabitat use appear to be far too evolutionarily labile to leave a persistent signature on morphological divergence across the phylogeny[73].

*Anolis* lizards in the Greater Antilles have apparently evolved extreme morphologies that are unique to islands. Specifically, the locomotor skeleton morphologies of grass-bush, crown-giant, and twig anoles are almost entirely absent from both mainland clades. This is consistent with previous studies of gross morphology[34,74] (but see ref. [32] for two instances of mainland convergence with grass-bush and crown-giant ecomorphs). The apparent lack of these specialized morphologies on the mainland may reflect that those niches were already occupied by other members of the rich continental lizard fauna, or possibly even other taxa. Alternatively, grass-bush, crown-giant, and twig ecomorphs might not be viable on the mainland because of, for example, high predation pressure[29]. Whatever the reason, the morphological diversification of the locomotor skeleton on the Greater Antilles relative to the Secondary Mainland clade fits the general prediction that evolution on islands can be faster, occur in bursts, and generate more extreme morphologies compared to mainland clades[75]. However, that the Primary Mainland has evolved equally disparate locomotor morphologies as the Greater Antillean *Anolis*, at even higher evolutionary rates, illustrates that pronounced diversification on islands is not a general pattern in *Anolis* (which is also evident from the limited diversification on the Lesser Antilles[76]).

In summary, the evolutionary modularity of the locomotor skeleton in Greater Antillean *Anolis* is consistent with adaptive change in response to ecological opportunity, while the persistence of this modularity in the clade that recolonized the mainland represents a significant phylogenetic inertia. Further investigating the developmental, functional, and ecological underpinnings of morphological variation is expected to yield valuable insights into how these aspects of evolution contribute toward phenotypic innovation and evolutionary change.

## Methods

**Micro-CT scanning.** We selected museum specimens based on a number of criteria, including completeness of the skeleton, sexual maturity, absence of malformations, and capture in the native range of the species. The selected museum specimens were scanned using microcomputed tomography (micro-CT) scanning using a GE phoenix v|tome|x m system (source voltage 100 kV, source current 200 μA, and isometric voxel size 55–75 μm) at the Nanoscale Facility of the University of Florida, US. Reconstructed image stacks (software GE phoenix datos|x CT) were further processed using VGStudio MAX software (version 3.2) by applying manual thresholding to extract surface models of skeletal structures.

**Quantification of morphology.** Linear measurements of limb bones were directly obtained using the VGStudio MAX software. We measured the maximum length of humerus, femur, ulna, tibia, and the individual phalangeal elements (including the claw) of the longest digit of both fore- and hindlimb (in mm to the closest 0.01 mm). We placed one point each on the proximal and on the distal end of the bone and extracted the distance between these two points in 3D space. For 47 bone elements (0.45% of the dataset), bones were fractured and no measurements were recorded. These missing values were imputed based on all linear measurements of all individuals using the pcaMethods R package[77] (version 1.78.0). All linear measurements were collected for one side (left or right) of each lizard. Measurements of lengths (including thresholding raw imaging data) were highly repeatable (Pearson's product-moment correlation $r = 0.992$, $P < 0.001$, $N = 40$). To generate an estimate of relative limb length (e.g., ref. [78]), all linear measurements of limb bones were divided by body size. We used centroid sizes of the pelvic girdles as a proxy of body size since it has been established that these centroid sizes are tightly correlated with snout-vent-length[73,79], a common measure of body size in lizards. Due to this strong correlation between centroid size and snout-vent-length, our dataset should be broadly comparable to the large body of literature on *Anolis* morphology that relies on snout-vent-length as an estimate of body size.

The shapes of the pectoral and pelvic girdles were quantified using landmark-based geometric morphometrics. After manual thresholding, meshes of segmented structures were exported in.stl format, which were converted into.ply format using the software MeshLab (version 2016.12)[80]. On left sides of the pectoral and the pelvic girdles, we placed 18 landmarks on informative anatomical features using the R package "geomorph" (version 3.1.3)[81]. Landmarks were developed by partially adopting published landmark sets for the pectoral[38] and pelvic girdles[39] and follow a previously described method[73]. When specimens showed damage on the left side, we landmarked the right sides of the structures and used the R package "StereoMorph"[82] (version 1.6.3) to mirror landmarks onto the left side. This was done for 31 pectoral girdles (4.40%) and 19 pelvic girdles (2.70%). Specimens lacking landmark data for pectoral or pelvic girdles were excluded from the analyses. All measurements of bone length and the placing of landmarks were performed blindly with respect to the identity of the specimen and by the same person. The repeatability of the landmarking procedure (including the thresholding of raw imaging data to extract mesh files), was assessed in a previous study and estimated to 0.98 for the pectoral girdle and 0.94 for the pelvic girdle[73]. We performed a "generalized Procrustes analysis" to obtain Procrustes shape variables (X-, Y-, and Z-coordinates) in the R package geomorph.

The resulting dataset capturing morphological variation in the locomotor skeleton contained 108 landmark-derived traits (each 18 Procrustes shape variables with X-, Y-, and Z-coordinates per pectoral and pelvic girdle), 15 traits capturing limb length, and one trait capturing body size, totaling 124 traits. Note that this data set is very different from the traits (e.g., total limb, head, and tail lengths) that form the basis for most inference on *Anolis* adaptive radiation and convergence (e.g., ref. [83]). The full dataset used in all analyses can be found in Supplementary Data 1.

**Transformation of morphological dataset.** The morphological dataset comprised Procrustes shape variables of pectoral and pelvic girdles[42,84], linear measurements of limb bones (relative to body size), and pelvic centroid size as a proxy for body size, and is therefore not on a commensurate scale. To ensure that the analyses of this dataset return meaningful and interpretable results, the raw values were standardized (z-transformed) such that each column (i.e., trait) was centered to zero and divided by the standard deviation. This transformation was performed prior to all analyses that focused on the entire locomotor skeleton. Since this z-transformation changes some properties of the dataset (e.g., eliminating variances of each trait), we performed analyses that are sensitive to these transformations (e.g., disparity analysis) on subsets of the data that contained only girdle-shape data, or only limb-length estimates. In each subset, traits were rescaled to a mean of zero, but the natural variation of the data was preserved. For a detailed discussion of statistical considerations of combined datasets, see refs. [40,42].

**Assignment of sex.** To exclude shape changes that are attributed to sexual dimorphism rather than species differences, we only included males in our analyses. This is necessary because the nature of the downstream analyses does not allow "controlling" for sex as is common practice in linear mixed models, for example. During the process of micro-CT scanning, sex was assigned to all individuals based on external morphology, and cross-validated with information provided in museum catalogs (if present). When assignment of sex was not possible or ambiguous, we used the shape of the pelvic girdle to corroborate sex identity. The rationale is that, due to reproductive functions that differ between males and females, sex differences should be most pronounced in the shape of the pelvic girdle. We used a linear discriminant analysis to classify individuals lacking sex assignment based on a training set of individuals with sex unambiguously assigned. We only accepted sex assignments that had a posterior probability of ≥0.8. The method was first validated on a large dataset consisting of 693 pelvic shape variables of 214 *Anolis* species for which sex assignments were available, and which was arbitrarily split into trainings (60%) and test (40%) sets. We found that 93.49% of all specimens were correctly assigned to their sex class, and we therefore deem this method appropriate for assigning sex to individuals with unknown sex identity.

**Species selection and phylogeny.** Since species status of some *Anolis* taxa is under debate, we included all *Anolis* taxa that are currently (December 2019) recognized as species by the Reptile Database[85]. Species were assigned to any of the biogeographic groups according to Poe et al.[26]: Greater Antilles, Lesser Antilles, Primary and Secondary Mainland. As outgroups, we included one representative species per genus of the Corytophanidae family which is the sister group to *Anolis* (*Basiliscus vittatus*, *Corytophanes cristatus*, and *Laemanctus longipes*) and *Polychrus gutturosus*, a distantly related, but anole-like lizard[86]. A list of specimens and the museum catalog numbers are provided in Supplementary Data 2.

The final dataset consisted of a total of 704 individuals of 271 species (2.60 individuals per species [ind./sp.]) with 110 species from the Greater Antillean (2.93 ind./sp.), 21 from the Lesser Antillean (2.81 ind./sp.), 29 from the Primary Mainland (2.14 ind./sp.), 107 from the Secondary Mainland group (2.34 ind./sp.), and 4 non-*Anolis* species (2.75 ind./sp.). Since we are interested in species-level comparisons, we averaged individual measurements between individuals per species.

A phylogenetic tree containing all species studied here was constructed by extending the maximum clade credibility (MCC) tree of Poe et al.[26] as follows. Ten *Anolis* species that were not included in this phylogeny were grafted onto the tree at positions suggested by the literature[87–91] with conservative branch lengths. Concerning the outgroup, the MCC tree published by Poe et al. contained one member each of the genus *Basiliscus* and *Polychrus*, and we grafted *C. cristatus* and *L. longipes* onto the tree at positions suggested by Pyron et al.[86] and with branch lengths adjusted according to divergence time estimates of the timetree.org[43] in proportion to the split between *Anolis* and Corytophanidae.

**Principal component analyses and patterns of variation**. We used principal component analysis (R package "stats") to gauge broad patterns of variation between all species in this dataset. We visualized these patterns by plotting the first two PCs in 2D overlain by a "phylomorphospace" constructed using the phytools R package (version 0.6-99)[92], and by plotting the first three PCs in 3D space using the rgl R package (version 0.100.30). We calculated the multivariate phylogenetic signal $K_{mult}$ of the full dataset and separately for limbs and girdles using the R package geomorph (version 3.1.3)[81]. The identity of the most average and most extreme *Anolis* species were determined by assessing each species' distance in PC scores (mean of PC1–PC3) from the mean of all *Anolis* species' PC scores. Morphological disparity was estimated separately for limbs and girdles by computing Procrustes variances for each of the groups in geomorph. The proportion of morphological variation explained by ecomorph classification was assessed using the function procD.pgls in the R package geomorph and was performed on a dataset and phylogenetic tree pruned to contain only species assigned to an ecomorph class[29,93].

**Evolutionary modularity and integration**. To quantify which modularity hypothesis was best supported by our data set (and subsets thereof), we estimated the CR[48] for each data set and each configuration. This measure describes the independence between supposed modules by contrasting the covariation *within* modules to that *between* modules. High independence (i.e., high modularity), results in low CR values (close to 0), whereas low independence (i.e., low modularity) results in a CR value approaching 1. We adopted the methodology proposed by Adams and Collyer[49] and derived z-score effect size that allowed us to compare the support for each modularity hypothesis between and within the groups. In brief, this approach uses a permutation approach (here, 1000 iterations) to derive an empirical null distribution against which the observed CR value is compared resulting in the CR z-score effect size ("compare.CR" function in geomorph). In addition, we used the R package EMMLi[94] (version 0.0.3) to evaluate the support for the different modularity hypotheses in a maximum likelihood framework (see Supplementary Table 14 and Supplementary Note 1).

Similarly, we assessed the strength of evolutionary integration in pairs of morphological blocks (or modules) by using PLS analyses in the R package geomorph. In brief, this approach uses a singular value decomposition of the covariance matrix between two blocks that describes the maximal covariation between them[95]. This is described by the first set of linear combinations (PLS1 vectors) in each of the two blocks. Scores per species projected onto these axes are used to estimate the maximum correlation $r_{PLS}$. To be able to compare these maximum correlations between groups, we adopted a methodology that is conceptually identical to the one used for modularity outlined above and derived PLS z-scores from permutations ("compare.pls" function in geomorph). Since we detected that the PLS method, and thus the derived PLS z-scores, is sensitive to the number of species included (Supplementary Fig. 2c), we adopted a subsampling strategy similar to Dellinger et al.[50] to equalize the proportion of species sampled per group. We derived 1000 subsampled data sets and summarized the differences in PLS z-scores between groups as the percentage of data sets that resulted in a significant *P* value (significance level 0.05).

**Evolutionary rates**. Net evolutionary rates of body size for each of the major groups[96] were computed in the R package geomorph. We estimated the occurrence of evolutionary rate shifts along distinct branches (edge shift) or entire clades (node shift) of the phylogenetic tree in a Bayesian framework using BayesTraits (version 3). In brief, a variable rate model was implemented and a rjMCMC approach was used to estimate the location, probability, and magnitude of rate shifts[51]. As input data, we used the principal components (PCs) rather than raw data since the orthogonality between PC axes fulfils the assumption of the model that traits are independent (see refs. [24,52,53] for similar approaches). PCs were derived from the full dataset, the full dataset without centroid size, or subsets (girdle or limb-length data), and we kept the number of PCs that were deemed explanatory by the 'broken stick' method as implemented in the R package vegan (version 2.5–6). To assess the impact of body size, which evolves fast in the Greater Antillean group (see above), we performed the same analyses on the full dataset but without centroid size. We performed two independent runs, each with default priors and 200 million iterations. The first 20% of iterations were removed as burn-in and a thinning factor of 100 was applied. We also ran the same models but with rates constrained to be equal across the tree and used Bayes factors calculated from the marginal likelihoods derived from a stepping stone sampler[97]. Bayes factors above 1300 for the entire locomotor skeleton and above 4 for individual blocks confirmed that the variable rate models had consistently higher support than the equal rate models

(Supplementary Table 15). The effective sample sizes reported by the software Tracer (version 1.5) were consistently above the recommended value of 200. We used the BTprocessR R package (version 0.0.1) to confirm that each chain had converged and to summarize their output. We identified edge as well as node shifts that occurred in the majority (>50%) of the posterior samples. The rationale for this threshold is that we were interested in comparing the tendency to vary in evolutionary rates between the major groups, not in identifying definite, highly supported rate shifts. We compared the variances in evolutionary rates between groups by comparing the homogeneity of variance. For this purpose, we extracted the mean, log-transformed, relative evolutionary rate of each branch from the posterior distribution and averaged this value between the two independent runs. Group identity was assigned to each branch using the "ace" function in the R package ape (version 5.3, Supplementary Fig. 6). We statistically assessed the equality of variances in evolutionary rates between major groups by applying Brown–Forsythe[98] tests. This test is a modified version of Levene's test[99] and is equivalent to a one-way ANOVA with the dependent variable being the absolute deviations from the group median. The Brown–Forsythe test therefore provides good robustness against non-normally distributed data and is less sensitive to outliers. In cases where the Brown–Forsythe test was significant (i.e., variances were found to be nonequal between the three major groups), we performed Tukey post hoc tests to identify significant differences between pairs of groups. Similarly, we assessed differences in relative evolutionary rates between the three major groups by applying Kruskal–Wallis tests[100]. In cases where this test was significant, we applied post hoc testing by performing multiple-comparison tests using the 'kruskalmc' function from the pgirmess R package (version 1.6.9).

**Reporting summary**. Further information on research design is available in the Nature Research Reporting Summary linked to this article.

## Data availability

Raw scans are available at Morphosource under the project name "Anolis sp.", project ID P1059 (https://www.morphosource.org/projects/0000C1059). Supplementary Data 1 contains the morphometric dataset used in the analyses. Supplementary Data 2 contains the digital object identifiers (DOIs) of the raw scan data for each individual used in this study.

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

## Acknowledgements

We are grateful to all curators and museum collection managers who loaned us specimens for this study with a special thanks to David C. Blackburn, and Coleman M. Sheehy III and Leroy Nunez for coordinating these loans. We thank Alex Pigot for sharing code for data visualization. We thank Rachel Blow, Cédric Aumont, and Maarten Vervoort for help with processing micro-CT data. This research was supported by a grant from the John Templeton Foundation (60501) to T.U., a grant from the Royal Physiographic Society of Lund to N.F., a Wenner–Gren postdoctoral fellowship to N.F., and a Wallenberg Academy Fellowship from the Knut and Alice Wallenberg to T.U.

## Author contributions

N.F., I.S.C.J., and T.U. conceived and coordinated the study. N.F. and I.S.C.J. collected data with assistance from E.L.S. N.F. and T.U. analyzed data with input from I.S.C.J. N.F. and T.U. wrote the paper. All authors commented on the draft.

## Funding

## Competing interests

The authors declare no competing interests.
