## [Peer Review File · Nature Communications]

Reviewers' Comments:

Reviewer #1:

Remarks to the Author:

Feiner et al

This is an interesting study with important comparisons.

The authors take on a challenge to test a hypothesis that is regularly stated in the literature yet rarely tested — that diversification rates in response to new habitats/niches are affected by certain inherited variational properties (meaning persistent patterns of covariation among body parts).

Using comparisons of anole radiations involving hundreds of species - radiations from mainland to island chain - to another island chain - back to mainland — they evaluate how patterns of covariation among body parts have changed as they moved from one set of ecological opportunities to another set (and so on) that changed the covariation patterns during lineage diversification.

The overall conclusion is that the interplay between ecological opportunity and phylogenetic inertia affects morphological diversification patterns. It is supported by the overall differences in the covariation of parts during the different radiations.

In step one - when anoles radiate from the (origin) Primary Mainland to the Greater Antilles islands, there is a change in ecological opportunities (I did wonder what exactly these different niches were and if they were quantified somewhere) and a co-incident change in body part modularity (parts that co-vary during diversification). However, while the modularity changed, the overall morphological variances in this comparison are about the same, and the rate of diversification and body size evolution are about the same between PM and GA. In other words, the Greater Antilles island colonizers were highly evolvable and were not “constrained” by ancestral modularity patterns. The authors don't address why the relatively high evolvability exists here, but it serves as a baseline for the further comparisons.

Then - step two - when they compared the Greater Antilles with the Secondary Mainland (a new radiation stemming from GA islands), the ecological opportunities changed again to mainland niches, but this time the body part modularity stayed the same as the ancestral island group, and here they observed that while the diversification rate was the same (i.e. just as many new species), the morphological variance was much less.

I believe they interpret these patterns to mean that the shift in modularity during the initial ‘islandification’ (changes associated with colonizing and radiating in the islands) of the Greater Antillian anoles might have led to the lack of ability of both the Secondary Mainland radiation and the Lesser Antilles radiation to diversify their morphology very much.

One of the other things that this study clarifies is the discovery that island radiations don't necessarily lead to more diverse morphologies than mainland:

“300 ...this Greater Antillean adaptive radiation was characterized by a strong co-evolution of limbs and

301 girdles, but that it did not produce more diverse morphologies than the clade that remained on 302 the mainland.”

In other words, the island habitats select for a remodularization of the limbs and girdles and still produced similar levels of morphological diversification.

Comments

This paper gives new perspective on an important evolutionary model system and aims to address an

important general concept in evolution about whether and how morphological diversification rates and extent are constrained by historical contingencies. The study succeeds in this aim and should be published.

Assuming the statistical analysis and the summary measures of rates and morphological disparity hold under scrutiny, I really cannot find a flaw in their basic comparative logic, but I have a couple comments about their further interpretation.

The role of developmental bias is not clear.

This is a complicated story, but there are clear alternate hypotheses. For example, the GA change in modularity and integration could be simple contingency in the colonizers? Would that change the conclusion?

"A role for developmental bias is supported by the fact that the clade that recolonized the 323 mainland exhibits an equally strong covariation between limbs and girdles as seen in the Greater 324 Antillean radiation. Moreover, while the Secondary Mainland clade produced an equal number of 325 species as the Greater Antillean clade (and more species than the Primary Mainland clade), the 326 morphologies of the locomotor skeleton are a subset of those that evolved on the Greater Antilles. 327 In contrast, the overlap with morphologies of the Primary Mainland clade is modest.

"Development" here seems to refer to phylogenetic inertia in covariation of modules. In fact, the nature of the covariation is unknown. Without knowledge of pleiotropy or regulatory connections, they need to be careful about invoking developmental bias. For example, the changes in modular covariation may be structural dependencies or pleiotropy or later stage functional integrations that are not developmental.

I don't think it's possible to distinguish these in this study.

I think this paper might need to add a simple schematic for the hypothesis connecting diversification rate, modularity, morphological variance, phylogeny.

Also "Ecology" is used here as the source of a functional adaptation to particular habitat. But anoles can adapt to the same habitat in different ways (Feiner et al 2020 eLife), making the possibilities more complex.

q- In reference to the unique morphologies in certain clades, how much do the conclusions depend on individual species existing in the dataset? Does one strange morph drive the conclusions?

The writing is a bit dense with terminology that might be defined in different ways, so they should take care to define terms.

Methods

They make a very rational categories of modularity hypotheses based on developmental genetic 'homology' and functional co-dependence of limb-girdle, or independence of all, or girdles together but limbs separate.

Species dataset seems balanced with distribution of species among locations

Reviewer #2:

Remarks to the Author:

I have carefully reviewed Feiner et al.'s manuscript, Evolution of the locomotor skeleton in Anolis lizards reflects the interplay between ecological opportunity and phylogenetic inertia. The manuscript

is interesting. In spite of years of work on *Anolis*, more papers comparing mainland and island anoles are truly needed to better understand the evolutionary dynamics of this well-studied clade. To that end, Feiner does a great job with their sampling (271 species!) While I can easily follow the logic of the manuscript, I have a series of critiques and suggestions for the authors that should be considered before publication.

1) Data archival: It is customary for CT scans and quantitative data to be archived for large studies such as this. Stanley is one of the organizers for a large collaborative project called, oVert which is archiving CT scan data from thousands of vertebrates. I would like to see a statement regarding the archival of these data before publication.

Along these same lines, a list of specimens and museums is critical to include in the supplementary materials.

2) Lns 54-57: These two sentences do not properly contrast ideas. Standing genetic variation still creates phenotypic variation via development. Development generates variation via heritable genetic variation. Consider rephrasing.

3) This manuscript advances a body of work that has begun in the early 1990s. This entire body of work has focused on size-corrected data and have used snout-to-vent length as a proxy for body size. This project is a significant deviation from that approach. To the best of my knowledge these authors do not size correct and analyze body size based on centroid size of the limbs and girdles. I think that the authors should caution readers that this methodological difference exists as it may account for differences between this study and others.

4) Ln. 100: "Certain differences" What differences?

5) The authors often refer to the "habitats" that the ectomorphs reside in, but "microhabitat" would be the more accurate term.

6) Lines 109-112 and Figure 1: I am really struggling to see how these clades are separated in any way on these PCs. Relative to the grey outgroup, all of the anoles seem piled right on top of one another. These statements can easily be removed without jeopardizing later analyses.

These PCs do not account for much variation, especially in morphological datasets where the first PC often accounts for 20-30% of the variation (or more). Furthermore, the loadings are all negligible, nothing over 0.19, when more would hold off on interpreting PCs of less than 0.5. It makes me feel like the authors are trying to squeeze water from a stone with the amount of space dedicated to this analysis. Interpreting patterns from this analysis seems tenuous.

First, I think that showing the scree plot of all PCs explaining more than 5% of variation is critical for readers to understand the significance of these PCs. Also, for the sake of the hypotheses being tested in this manuscript, I believe that only panels A, C, and E are necessary and useful. In spite of much effort, I cannot interpret panel B.

Finally, CT data are beautiful. Why not show readers the girdles and limbs with landmarks illustrated? This would be far more useful than the list of landmarks found in the SI.

7) Ln 118: "This?" The previous sentence is about phylogenetic signal. This sentence is about disparity. How does one follow from the others?

8) Ln 119-122: I cannot understand the values in the parentheses here. What are comparisons are the P-values referring to?

9) Lines 190-195 and Figure 2: More information is desperately needed here regarding the CR analysis. There is information hidden in the methods and figure legend, but that isn't enough. Most important, I have no idea how to interpret Fig. 2. How do the authors decide what model has the most explanatory power? Is it the lowest z-score for all models for each group? Can we only more narrowly compare the three lines among clades within each model?

10) Figure 3: Colors. The colors here blend together to the point where I cannot see the differences among the lineages or where colors change.

11) Discussion: These authors are not the first to describe phylogenetic signal in modularity and integration in anoles. Sanger and Losos have several papers arguing many of the same ideas, albeit only among island anoles. Those ideas may be worth considering here.

12) Discussion: The authors are missing an important step between morphology and microhabitat: performance. Is anything known about the way anoles or their relatives use their limbs differently? For example, are their differences in posture that may explain the differences in the girdle? Even if not, I believe that explicitly addressing the need for performance data would greatly strengthen the Discussion.

Reviewer #3:

Remarks to the Author:

This manuscript presents an analysis of evolutionary modularity in skeletal morphology of *Anolis* lizards from the Greater Antilles, Lesser Antilles, and mainland Central and South America. The paper is generally well written and clearly organized. The dataset is fairly large, including 704 individuals from 271 species (about 2/3 of *Anolis* species), and the analyses appear to be thorough (although I am not an expert in all of these methods).

I would classify the study as primarily descriptive and taxon specific, though. It's just not clear how much we learn from this study about evolutionary modularity or adaptive radiation more generally, at least as it's currently presented. Rather, the findings represent a more incremental increase in our knowledge about the *Anolis* radiations. For example, the results about the morphospace occupied by different clades (represented in Fig 1), although more detailed because of the CT scan data, are largely consistent with analyses based on traditional specimen measurements. And the finding of increased variability in evolutionary rates for some traits in some clades relative to others is also not meaningfully different from previous work. The finding of support for different modularity hypotheses between mainland and island clades is original, I think, but it's not clear whether these differences contributed to the adaptive radiations (or ecomorphological evolution) in islands vs mainland or whether they are in line with differences expected to be found in some subclades but not others due to chance or other evolutionary scenarios.

Minor Comments:

1) The figures do not translate to grayscale very well. The size of the branches in the circular tree in Fig 3 and the plots (as well as their labeling) makes this figure very difficult to evaluate.

2) I think the results represented in Fig 2 would be more informative as a table; i.e. it would be nice to see the actual numerical z-scores.

3) L322-333 (particularly "Yet, the results are intelligible if evolution on the Greater Antilles had persistent effects on the lizards' development and behavioural biology, thereby imposing a bias on their future evolution following re-colonization of the mainland."): The pattern in the secondary mainland clade isn't necessarily evidence for this, is it? i.e. We'd expect the Greater Antillean and secondary mainland clade to exhibit similar patterns simply because the secondary mainland clade

was derived from the Greater Antillean clade.

4) L374-374 ("This macroevolutionary pattern of diversification of the locomotor skeleton of *Anolis* reflects the reciprocal relationship between development and ecology in evolution."): It's not clear what's meant by this; it seems too broad and overly vague to be meaningful.

Reviewer #4:

Remarks to the Author:

I really like this paper. The authors have built a substantial dataset, following the basic methodology of Tininus & Russell (2014) and Tininus et al. (2018) but increasing the scale considerably to cover the majority of extant *Anolis* species. This enabled evaluation of any differences between the multiple diversifications of *Anolis* on the mainland and on the Antilles. A variety of statistical techniques are utilised by the authors for this investigation, all of which are justified in the text. Where necessary, reasonable steps were taken to account for potential sources of error. The conclusion drawn of a change in phylogenetic covariance of girdles and limbs between the original mainland clade and the Greater Antillean clade is well-supported by the data. The conclusion's significance is illustrated by the similarities between the divergences of the Greater Antillean and Secondary Mainland clades. I have a few minor notes, which are included below, but overall this is a very thorough study worthy of publication.

- A 3D model of the pelvic and pectoral girdles with the landmarks mapped on would be helpful to include as a supplementary figure, to improve the repeatability of the study. Tininus & Russell (2014) included a figure for their pelvic girdle study, so this isn't essential. However, I don't find their figure especially clear and it's generally good practice to include a figure to supplement the list of landmarks.
- Regarding the PCA, how much variance was explained by principal components subsequent to PC3? There's very little difference between the values for PC2 and PC3 and the first three principal components account for ~36% of the total observed variance. If other components also explain close to 10%, it would be nice to list them in a table in the supplementary, or possibly have supplementary figures showing PC1/PC4 etc.
- There's a slight error in the text in line 149: 20 species are listed for the combined Lesser Antillean clade, but there are 21 species in the figure and dataset.
- In line 102, the wording is unclear regarding which hypothesis is H1 and which is H2. From the phrasing of the sentence, I would conclude that H1 has the pelvic and pectoral girdles forming a single evolutionary module. However, in Figure 2 H1 is clearly labelled as the hypothesis in which the pelvic and pectoral girdles are treated as separate. Could you edit this to keep it consistent?
- I suggest rephrasing the sentence beginning on line 332 – maybe end it with "despite the genetic change being random" instead of the current phrasing.
- In line 362: are there any species that you think may be candidates to have occupied those niches on the mainland? This isn't essential to your point, but it could be interesting.
- In supplementary tables 3 & 4, it might be helpful to include some method to visually differentiate between the different values being reported. For example, minor shading of the Procustes variances would make it easier to interpret quickly.

Below is the report of our manuscript revision. We show sentences taken from the manuscript in italic, and parts newly inserted into the manuscript as underlined. Page and line numbers refer to positions in the document with tracked changes.

Reviewer #1 (Remarks to the Author):

This is an interesting study with important comparisons. The authors take on a challenge to test a hypothesis that is regularly stated in the literature yet rarely tested — that diversification rates in response to new habitats/niches are affected by certain inherited variational properties (meaning persistent patterns of covariation among body parts).

Using comparisons of anole radiations involving hundreds of species - radiations from mainland to island chain - to another island chain - back to mainland — they evaluate how patterns of covariation among body parts have changed as they moved from one set of ecological opportunities to another set (and so on) that changed the covariation patterns during lineage diversification.

The overall conclusion is that the interplay between ecological opportunity and phylogenetic inertia affects morphological diversification patterns. It is supported by the overall differences in the covariation of parts during the different radiations.

In step one - when anoles radiate from the (origin) Primary Mainland to the Greater Antilles islands, there is a change in ecological opportunities (I did wonder what exactly these different niches were and if they were quantified somewhere) and a co-incident change in body part modularity (parts that co-vary during diversification). However, while the modularity changed, the overall morphological variances in this comparison are about the same, and the rate of diversification and body size evolution are about the same between PM and GA. In other words, the Greater Antilles island colonizers were highly evolvable and were not “constrained” by ancestral modularity patterns. The authors don’t address why the relatively high evolvability exists here, but it serves as a baseline for the further comparisons.

Then - step two - when they compared the Greater Antilles with the Secondary Mainland (a new radiation stemming from GA islands), the ecological opportunities changed again to mainland niches, but this time the body part modularity stayed the same as the ancestral island group, and here they observed that while the diversification rate was the same (i.e. just as many new species), the morphological variance was much less.

I believe they interpret these patterns to mean that the shift in modularity during the initial ‘islandification’ (changes associated with colonizing and radiating in the islands) of the Greater Antillian anoles might have led to the lack of ability of both the Secondary Mainland radiation and the Lesser Antilles radiation to diversify their morphology very much.

One of the other things that this study clarifies is the discovery that island radiations don’t necessarily lead to more diverse morphologies than mainland: “300 ...this Greater Antillean adaptive radiation was characterized by a strong co-evolution of limbs and 301 girdles, but that it did not produce more diverse morphologies than the clade that remained on 302 the mainland.” In other words, the island habitats select for a remodularization of the limbs and girdles and still produced similar levels of morphological diversification.

Response: We thank the reviewer for the succinct summary of our study and for emphasizing the relevance of our study.

Comments

This paper gives new perspective on an important evolutionary model system and aims to address an important general concept in evolution about whether and how morphological diversification rates and extent are constrained by historical contingencies. The study succeeds in this aim and should be published.

Assuming the statistical analysis and the summary measures of rates and morphological disparity hold under scrutiny, I really cannot find a flaw in their basic comparative logic, but I have a couple comments about their further interpretation.

The role of developmental bias is not clear. This is a complicated story, but there are clear alternate hypotheses. For example, the GA change in modularity and integration could be simple contingency in the colonizers? Would that change the conclusion?

Response: The reviewer is correct in that these data cannot conclusively establish the reasons for changes in modularity structure following the colonization of the Greater Antilles. This does not, however, affect the main conclusion that the adaptive radiation proceeded along a different modularity structure relative to the Primary Mainland clade and that this structures was retained in the clade that re-colonized the mainland. To emphasize that our results are ‘consistent with’, but do not prove, the ecological hypothesis, we have rephrased the relevant sentence in the Discussion.

Originally: *One possible explanation is that Greater Antillean species are adapted to ecological conditions that are peculiar to islands.*

Revised (lines 271-273): *While the reasons for this shift cannot be confidently established from these data, one possible explanation is that Greater Antillean species are adapted to ecological conditions that are peculiar to islands.*

“A role for developmental bias is supported by the fact that the clade that recolonized the 323 mainland exhibits an equally strong covariation between limbs and girdles as seen in the Greater 324 Antillean radiation. Moreover, while the Secondary Mainland clade produced an equal number of 325 species as the Greater Antillean clade (and more species than the Primary Mainland clade), the 326 morphologies of the locomotor skeleton are a subset of those that evolved on the Greater Antilles. 327 In contrast, the overlap with morphologies of the Primary Mainland clade is modest.

“Development” here seems to refer to phylogenetic inertia in covariation of modules. In fact, the nature of the covariation is unknown. Without knowledge of pleiotropy or regulatory connections, they need to be careful about invoking developmental bias. For example, the changes in modular covariation may be structural dependencies or pleiotropy or later stage functional integrations that are not developmental. I don’t think it’s possible to distinguish these in this study.

Response: We believe that the underlying reason for this comment are divergent views on what ‘development’ encompasses. In our opinion, pleiotropy and regulatory interactions are clearly part of development, as are integrations that result from, for example, mechanical interaction between parts (see e.g., the textbook *Developmental Biology* by Gilbert & Barresi or the textbook *Mechanisms of Morphogenesis* by Davies). It is possible that the relevant mechanisms play out

after hatching (i.e., after embryogenesis), but the term ‘developmental bias’ is not limited to changes that occur in embryology (following reviews by Maynard-Smith et al. 1985; Uller et al. 2018). Nevertheless, we have made several changes to the manuscript (Abstract and Discussion) to reduce the risk for misunderstandings.

Original: *We suggest that these macroevolutionary trends of the locomotor skeleton in *Anolis* illustrate that adaptation to ecological opportunities on islands can have profound effects on development, with lasting effects on patterns of morphological diversification.*

Revised (lines 42-45): *We suggest that these macroevolutionary trends of the locomotor skeleton in *Anolis* illustrate that adaptation to ecological opportunities on islands can have profound effects on trait development, with lasting effects on patterns of morphological diversification.*

Original: *It is therefore plausible that the distinct evolutionary modularity of the Greater Antillean clade, relative to the Primary Mainland clade, in part reflects an ancient and persistent difference in the developmental integration of girdles and limbs.*

Revised (lines 278-281): *It is therefore plausible that the distinct evolutionary modularity of the Greater Antillean clade, relative to the Primary Mainland clade, in part reflects an ancient and persistent difference in how girdles and limbs develop together (‘developmental bias’¹³).*

I think this paper might need to add a simple schematic for the hypothesis connecting diversification rate, modularity, morphological variance, phylogeny.

Response: We thank the reviewer for this suggestion and have played with some options. Unfortunately, we believe that it is not easy to capture the hypothesis in a simple schematic, and this schematic would still require the reader to work through the text to not risk being misleading. We have therefore not added another figure to the paper.

Also “Ecology” is used here as the source of a functional adaptation to particular habitat. But anoles can adapt to the same habitat in different ways (Feiner et al 2020 eLife), making the possibilities more complex.

Response: Yes, it is true that there is no simple one-to-one mapping of morphology and function, but this does not preclude that adaptation can lead to a change in patterns of covariation, or that this covariation can channel diversification along particular trajectories. Nevertheless, we agree that a better understanding of the functional interactions between morphology and ecology in *Anolis* would be very useful and we have included this point in our revised manuscript (see our response to the last comment of Reviewer #2).

q- In reference to the unique morphologies in certain clades, how much do the conclusions depend on individual species existing in the dataset? Does one strange morph drive the conclusions?

Response: This is an important consideration and we thank the reviewer for bringing this up. We have tested if our main conclusion about the differences in modularity structure is robust to the exclusion of the most extreme morphologies in the Greater Antillean clade. We find that the H3

modularity structure of this modified dataset again received a significantly higher support than all other modularity structures, indicating that exaggerated morphologies are not driving this result. We have added the new result of this additional analysis as Supplementary Table 9 and added one sentence to the Results section.

Sentence added (lines 184-186): *This results persisted if the 10% of the species with the most extreme morphologies of the Greater Antillean clade are excluded, which demonstrated that the modularity structure is not driven by exaggerated morphologies (Supplementary Table 9).*

The writing is a bit dense with terminology that might be defined in different ways, so they should take care to define terms.

Response: We are sympathetic to this comment and have tried to eliminate ambiguity by spelling out what we mean in plain terms, as explained above for ‘development’. While definitions can resolve ambiguity, the text would become very dense with too many definitions, and we therefore generally tried to explain terms in more general language upon first mentioning, leaving technical detail for the methods (e.g., explaining evolutionary modularity as ‘the extent to which parts of the organism evolve together’ and ‘covariation of traits across a phylogeny’ in the main text). Several examples of how we have revised our manuscript with this in mind can be found in the response to specific comments above and below. Further examples include the following:

Original: *It is therefore plausible that the distinct evolutionary modularity of the Greater Antillean clade, relative to the Primary Mainland clade, in part reflects an ancient and persistent difference in the developmental integration of girdles and limbs.*

Revised (lines 278-281): *It is therefore plausible that the distinct evolutionary modularity of the Greater Antillean clade, relative to the Primary Mainland clade, in part reflects an ancient and persistent difference in how girdles and limbs develop together (‘developmental bias’¹³).*

Originally: *For example, the peculiar neonatal biology of marsupials, [...]*

Revised (lines 283, 284): *For example, the peculiar reproductive biology of marsupials, [...]*

Methods

They make a very rational categories of modularity hypotheses based on developmental genetic ‘homology’ and functional co-dependence of limb-girdle, or independence of all, or girdles together but limbs separate.

Species dataset seems balanced with distribution of species among locations.

Response: Thank you for this positive evaluation of our methodology.

Reviewer #2 (Remarks to the Author):

I have carefully reviewed Feiner et al.’s manuscript, Evolution of the locomotor skeleton in Anolis lizards reflects the interplay between ecological opportunity and phylogenetic inertia. The manuscript is

interesting. In spite of years of work on *Anolis*, more papers comparing mainland and island anoles are truly needed to better understand the evolutionary dynamics of this well-studied clade. To that end, Feiner does a great job with their sampling (271 species!) While I can easily follow the logic of the manuscript, I have a series of critiques and suggestions for the authors that should be considered before publication.

Response: We are grateful for the positive assessment of our work and the reviewer for his/her time.

1) Data archival: It is customary for CT scans and quantitative data to be archived for large studies such as this. Stanley is one of the organizers for a large collaborative project called, oVert which is archiving CT scan data from thousands of vertebrates. I would like to see a statement regarding the archival of these data before publication.

Response: We agree with the reviewer that deposition of CT scans in publicly available databases is absolutely essential. We have archived our data on MorphoSource, which is the repository that is also used by oVert. Within MorphoSource, each specimen is linked to its iDigBio, which further improves the accessibility to researchers interested in using our scan data. In addition, we provide source data file 2 which contains the digital object identifiers (DOIs) and the museum voucher IDs of each specimen and other metadata. The section 'Data availability' contains the information on how this data can be accessed (lines 524-528):

"Raw scans are available at Morphosource under the project name 'Anolis sp.', project ID P1059. Source data file 1 contains the morphometric dataset used in the analyses. Source data file 2 contains the digital object identifiers (DOIs) of the raw scan data for each individual used in this study."

Along these same lines, a list of specimens and museums is critical to include in the supplementary materials.

Response: We fully agree. We have now included source data file 2 that contains the necessary information.

2) Lns 54-57: These two sentences do not properly contrast ideas. Standing genetic variation still creates phenotypic variation via development. Development generates variation via heritable genetic variation. Consider rephrasing.

Response: This is a good point and we have rephrased accordingly.

Originally: *In the short term, the capacity to adapt is limited by standing genetic variation^{6,7}.*

Revised (lines 54-56): *In the short term, the capacity to adapt is limited by the phenotypic variation generated by standing genetic variation^{6,7}.*

3) This manuscript advances a body of work that has begun in the early 1990s. This entire body of work has focused on size-corrected data and have used snout-to-vent length as a proxy for body size. This

project is a significant deviation from that approach. To the best of my knowledge these authors do not size correct and analyze body size based on centroid size of the limbs and girdles. I think that the authors should caution readers that this methodological difference exists as it may account for differences between this study and others.

Response: It is correct that our methodology based on geometric morphometrics differs from more traditional work on *Anolis* that uses univariate measures to quantify morphology. Regarding the size correction, Tinius (2016) has systematically explored the correlation between snout-vent-length and centroid size in Iguanid lizards and found a tight correlation ($r = 0.98$, $P\text{-value} < 0.001$). We have further explored this relationship within two *Anolis* species (Feiner *et al.*, 2020, *eLife*) and found a similarly tight correlation (*A. carolinensis*: $r = 0.96$, $P\text{-value} < 0.001$, $N = 45$; *A. sagrei*: $r = 0.97$, $P\text{-value} < 0.001$, $N = 71$). We therefore do not expect that using centroid size rather than snout-vent-length had any impact on the conclusions presented in our paper. Nevertheless, we have added a sentence to alert the reader to this difference.

Sentence added (lines 375-378): *Due to this strong correlation between centroid size and snout-vent-length, our dataset should be broadly comparable to the large body of literature on Anolis morphology that rely on snout-vent-length as an estimate of body size.*

4) Ln. 100: “Certain differences” What differences?

Response: We have added more detail in this sentence to specify how the z-transformation affects the data structure.

Original: *This procedure removes certain properties from the dataset^{39,40}, but the transformation allows us to infer morphological differences among Anolis clades across the entire locomotor skeleton, which is our primary focus.*

Revised (lines 101-104): *This procedure removes certain properties (e.g., the original trait variances) from the dataset^{39,40}, but the transformation allows us to infer morphological differences among Anolis clades across the entire locomotor skeleton, which is our primary focus.*

5) The authors often refer to the “habitats” that the ectomorphs reside in, but “microhabitat” would be the more accurate term.

Response: We agree and have modified the term where applicable (lines 134 and 135).

6) Lines 109-112 and Figure 1: I am really struggling to see how these clades are separated in any way on these PCs. Relative to the grey outgroup, all of the anoles seem piled right on top of one another. These statements can easily be removed without jeopardizing later analyses.

Response: We agree that the differences are rather small and have therefore removed the descriptive part that is not relevant for our argument.

Original: *Anolis species belonging to the Greater Antilles, Lesser Antilles and the Primary and Secondary Mainland clades occupy largely overlapping regions in morphospace, although species inhabiting the Lesser Antilles tend to score low on the first principal*

component (PC1), and the Primary Mainland and Greater Antillean species tend to separate along the second and third PCs (Fig. 1b and c, and Extended Data Fig. 1).

Revised (lines 110-114): *Anolis species belonging to the Greater Antilles, Lesser Antilles and the Primary and Secondary Mainland clades occupy largely overlapping regions in morphospace, although slight differences exist (Fig. 1b and c, and Extended Data Fig. 2).*

These PCs do not account for much variation, especially in morphological datasets where the first PC often accounts for 20-30% of the variation (or more). Furthermore, the loadings are all negligible, nothing over 0.19, when more would hold off on interpreting PCs of less than 0.5. It makes me feel like the authors are trying to squeeze water from a stone with the amount of space dedicated to this analysis. Interpreting patterns from this analysis seems tenuous.

Response: First, we agree that variation seems to be distributed fairly evenly among our 124 traits and that PC1 explains comparably little variation. Nevertheless, the first three PCs are able to discriminate between ecomorphs (see Extended Data Figure 2), which shows that they are picking up meaningful variation.

Nevertheless, we have followed the reviewer's advice and toned down our interpretation of the morphospace differences among clades (see our response above). None of our conclusions depends on these. We keep the Extended Data Fig. 2 so that readers can explore the differences in morphospace occupancy between the clades and ecomorphs, as this is likely of interest to many researchers.

First, I think that showing the scree plot of all PCs explaining more than 5% of variation is critical for readers to understand the significance of these PCs. Also, for the sake of the hypotheses being tested in this manuscript, I believe that only panels A, C, and E are necessary and useful. In spite of much effort, I cannot interpret panel B.

Response: First, we have added a scree plot to provide the reader with all relevant information about our analyses and also added principal component plots of the five PCs that explain more than 5% of variance (see modified Extended Data Fig. 2). Second, we regret that the reviewer does not see value in panels b, d and f of Figure 1. We believe that they convey important information that would be missing otherwise. Most importantly, by comparing panels c to d, and e to f, one can deduce which ecomorphs that exhibit variation that is absent from clades outside the Greater Antilles (i.e., grass-bush, crown-giant and twig). We believe that this information would be lost if the panels d and f were removed.

Finally, CT data are beautiful. Why not show readers the girdles and limbs with landmarks illustrated? This would be far more useful than the list of landmarks found in the SI.

Response: We agree with this suggestion and have added a new Extended Data Fig. 1 to illustrate the placement of the landmarks and the positioning of the univariate measurements taken on fore- and hindlimbs. In combination with the exact anatomical description of the landmarks in Supplementary Table 1, we believe that this provides detailed information on our methodology that will be a useful resource for other researchers.

7) Ln 118: “This?” The previous sentence is about phylogenetic signal. This sentence is about disparity. How does one follow from the others?

Response: This sentence was phrased ambiguously; we have modified it.

Originally: *This demonstrated that species of the Greater Antilles occupy a larger volume in morphospace [...]*

Revised (lines 120, 121): *Using this framework, we find that species of the Greater Antilles occupy a larger volume in morphospace [...]*

8) Ln 119-122: I cannot understand the values in the parentheses here. What are comparisons are the P-values referring to?

Response: The contrasts are between Procrustes Variances of different clades. We have added a subscript to each P-value that indicated which contrast it is referring to (e.g., $P_{girdles_GAVsLA}$ instead of $P_{girdles}$; see lines 124-128). This should enhance the readability.

9) Lines 190-195 and Figure 2: More information is desperately needed here regarding the CR analysis. There is information hidden in the methods and figure legend, but that isn't enough. Most important, I have no idea how to interpret Fig. 2. How do the authors decide what model has the most explanatory power? Is it the lowest z-score for all models for each group? Can we only more narrowly compare the three lines among clades within each model?

Response: To improve the clarity of this figure, we have restructured it so that all z-scores of one clade are shown next to each other. Although the nature of z-scores (effect sizes) makes it possible to compare both within and across clades, our primary focus here lies on the within-clade comparison. The visual representation now better reflects this, and it should help the reader to grasp and interpret the results. We have also added information on how to interpret which model is the best (i.e., lowest CR z-score) to the legend of Figure 2 and an additional sentence about the methodology.

Sentence added (lines 808-810): *CR z-scores are derived from a permutation approach (here, 1000 iterations) to derive an empirical null distribution against which the observed CR value of each configuration is compared (see Methods for more details).*

Original: *The modularity hypothesis H1, limbs forming a single evolutionary module, had the highest support in the Primary Mainland clade.*

Revised (lines 811, 812): *The modularity hypothesis H1, limbs forming a single evolutionary module, had the highest support (i.e., lowest CR z-score) in the Primary Mainland clade.*

Original: *[...] had a significantly higher support than the other hypotheses in the Greater Antillean and the Secondary Mainland clade.*

Revised (lines 813-815): *[...] had a significantly higher support (i.e., lower CR z-score) than the other hypotheses in the Greater Antillean and the Secondary Mainland clade.*

10) Figure 3: Colors. The colors here blend together to the point where I cannot see the differences among the lineages or where colors change.

Response: We apologize for the unsatisfactory quality of the image files. At the initial submission stage, figures were embedded in the manuscript file, which negatively affected the quality (resolution) of the figures. We have now supplied figures separately and in final (print) quality. We believe that this resolves the issue.

11) Discussion: These authors are not the first to describe phylogenetic signal in modularity and integration in anoles. Sanger and Losos have several papers arguing many of the same ideas, albeit only among island anoles. Those ideas may be worth considering here.

Response: We thank the reviewer for suggesting to include a link to this literature and have ensured that this literature is mentioned at two relevant points in the Discussion section.

Sentence added (lines 304-307): *Similar mainland and island comparisons of other defining features of the Greater Antillean adaptive radiation, such as the morphology of skull, would be highly interesting.*

Original: *In Anolis, it would be particularly interesting to compare morphological variation within species of the three major clades.*

Revised (lines 314-418): *For example, the within-species integration of the skull can vary between ecomorphs in Greater Antillean Anolis⁶⁴, but how these features of the skull co-evolve across the phylogeny remains to be explored. With respect to the locomotor skeleton, it would be particularly interesting to compare patterns of morphological variation within species from each of the three major clades.*

Reference added: *64. Sanger, T. J., Mahler, D. L., Abzhanov, A. & Losos, J. B. Roles for modularity and constraint in the evolution of cranial diversity among Anolis lizards. Evolution 66, 1525-1542 (2012).*

12) Discussion: The authors are missing an important step between morphology and microhabitat: performance. Is anything known about the way anoles or their relatives use their limbs differently? For example, are their differences in posture that may explain the differences in the girdle? Even if not, I believe that explicitly addressing the need for performance data would greatly strengthen the Discussion.

Response: The reviewer raises an important point here. While the relationship between (relative) limb length and climbing and running performance is relatively well researched, the impact of girdle morphology on performance is uncharted scientific territory. We briefly mentioned this gap in knowledge in our original Discussion, but we have taken this opportunity to make this point more explicit.

Originally: *Lizards from island and mainland clades may also interact differently with their environment^{32-34,73}, but plastic responses to different microhabitat use appears to be far too evolutionarily labile to leave a persistent signature on morphological divergence across the phylogeny.*

Revised (lines 323-328): *Lizards from island and mainland clades may also interact differently with their environment, and more information on functional aspects of morphology, including how morphological differences impact on ecological performances, are sorely needed^{32-34,71}. However, plastic responses to different microhabitat use appears to be far too evolutionarily labile to leave a persistent signature on morphological divergence across the phylogeny⁷².*

Reviewer #3 (Remarks to the Author):

This manuscript presents an analysis of evolutionary modularity in skeletal morphology of *Anolis* lizards from the Greater Antilles, Lesser Antilles, and mainland Central and South America. The paper is generally well written and clearly organized. The dataset is fairly large, including 704 individuals from 271 species (about 2/3 of *Anolis* species), and the analyses appear to be thorough (although I am not an expert in all of these methods).

I would classify the study as primarily descriptive and taxon specific, though. It's just not clear how much we learn from this study about evolutionary modularity or adaptive radiation more generally, at least as it's currently presented. Rather, the findings represent a more incremental increase in our knowledge about the *Anolis* radiations. For example, the results about the morphospace occupied by different clades (represented in Fig 1), although more detailed because of the CT scan data, are largely consistent with analyses based on traditional specimen measurements. And the finding of increased variability in evolutionary rates for some traits in some clades relative to others is also not meaningfully different from previous work. The finding of support for different modularity hypotheses between mainland and island clades is original, I think, but it's not clear whether these differences contributed to the adaptive radiations (or ecomorphological evolution) in islands vs mainland or whether they are in line with differences expected to be found in some subclades but not others due to chance or other evolutionary scenarios.

Response: We appreciate the reviewer's critical evaluation of our study. However, we wish to make three brief counter-arguments. Firstly, it is not possible to resolve the issues we address here without studying a well-defined clade of organisms, densely sampled across the phylogeny (e.g., adopting a patchy sampling of all iguanids would have been inappropriate for our purposes). That a study makes use of a specific group of organisms does not necessarily mean that its findings are 'taxon-specific'. Secondly, although we confined ourselves to one genus (*Anolis*), our study addresses a classical macroevolutionary problem, namely patterns of diversification of mainland versus island clades. The evolutionary history of *Anolis* with repeated shifts between mainland and islands makes them a highly appropriate choice to address this question. Thirdly, before this work, almost all studies on the ecomorphological evolution in *Anolis* have relied on a single univariate measure (i.e., limb length). While we recover some of the same results (e.g., most ecomorphs can be distinguished), it is only by studying covariation between limb bones and the shape of girdles that we are able to address the core questions: the evolution of modularity, integration, and whether or not there is evidence for a persistent developmental bias in adaptive diversification.

We note that these strengths of our study were noted and appreciated by the other reviewers, and we have therefore refrained from boosting them further in the Introduction or Discussion of the revised manuscript.

Minor Comments:

1) The figures do not translate to grayscale very well. The size of the branches in the circular tree in Fig 3 and the plots (as well as their labeling) makes this figure very difficult to evaluate.

Response: A similar point was raised by Reviewer # 2 and we refer to our response above.

2) I think the results represented in Fig 2 would be more informative as a table; i.e. it would be nice to see the actual numerical z-scores.

Response: To present the results in Fig. 2 more clearly, we have restructured this figure (see response to Reviewer #2 above). The full output of the statistical models (including numerical z-scores) can be found in Supplementary Tables 6-8. To make this information more prominent in our manuscript, we have added a sentence to the legend of Fig. 2.

Sentence added (lines 815, 816): *For statistical support of alternative models, see Supplementary Tables 6-8.*

3) L322-333 (particularly "Yet, the results are intelligible if evolution on the Greater Antilles had persistent effects on the lizards' development and behavioural biology, thereby imposing a bias on their future evolution following re-colonization of the mainland."): The pattern in the secondary mainland clade isn't necessarily evidence for this, is it? i.e. We'd expect the Greater Antillean and secondary mainland clade to exhibit similar patterns simply because the secondary mainland clade was derived from the Greater Antillean clade.

Response: Yes, we would expect the Greater Antilles and Secondary Mainland species to evolve in a similar manner if there was a persistent developmental bias. This is indeed how we interpret the results (see lines 302-304). But please note that this result is unexpected from a perspective of natural selection on unbiased variation: if this were the case, ecological differences between islands and mainland would have driven the adaptive radiation, and hence the two mainland groups would have evolved more similar to each other than to the radiation on the Greater Antilles.

4) L374-374 ("This macroevolutionary pattern of diversification of the locomotor skeleton of Anolis reflects the reciprocal relationship between development and ecology in evolution."): It's not clear what's meant by this; it seems too broad and overly vague to be meaningful.

Response: We agree with this criticism and have removed this sentence (lines 348, 349).

Reviewer #4 (Remarks to the Author):

I really like this paper. The authors have built a substantial dataset, following the basic methodology of

Tinius & Russell (2014) and Tinius et al. (2018) but increasing the scale considerably to cover the majority of extant *Anolis* species. This enabled evaluation of any differences between the multiple diversifications of *Anolis* on the mainland and on the Antilles. A variety of statistical techniques are utilised by the authors for this investigation, all of which are justified in the text. Where necessary, reasonable steps were taken to account for potential sources of error. The conclusion drawn of a change in phylogenetic covariance of girdles and limbs between the original mainland clade and the Greater Antillean clade is well-supported by the data. The conclusion's significance is illustrated by the similarities between the divergences of the Greater Antillean and Secondary Mainland clades. I have a few minor notes, which are included below, but overall this is a very thorough study worthy of publication.

Response: We are grateful for the strong endorsement of our study and thank the reviewer for providing us with constructive feedback.

- A 3D model of the pelvic and pectoral girdles with the landmarks mapped on would be helpful to include as a supplementary figure, to improve the repeatability of the study. Tinius & Russell (2014) included a figure for their pelvic girdle study, so this isn't essential. However, I don't find their figure especially clear and it's generally good practice to include a figure to supplement the list of landmarks.

Response: A similar point was raised by Reviewer #2 and we have added Extended Data Fig. 1 to close this gap (see above).

- Regarding the PCA, how much variance was explained by principal components subsequent to PC3? There's very little difference between the values for PC2 and PC3 and the first three principal components account for ~36% of the total observed variance. If other components also explain close to 10%, it would be nice to list them in a table in the supplementary, or possibly have supplementary figures showing PC1/PC4 etc.

Response: We agree that this information was previously missing from our manuscript. To provide the reader with a more complete overview of the results of our principal component analyses, we have added more information to the Extended Data Figure 2. Please see our response to a similar comment made by Reviewer #2 above for a more detailed explanation.

- There's a slight error in the text in line 149: 20 species are listed for the combined Lesser Antillean clade, but there are 21 species in the figure and dataset.

Response: Thank you for spotting this inconsistency. The difference between the total number of species (21) and the sum of species in the two Lesser Antillean clades (9 and 11) is explained by a single species (*A. acutus*) which represents an additional colonization of the Lesser Antilles from the Greater Antilles. Since the Lesser Antillean species are not central to our arguments and to avoid unnecessarily dense information in the Results section, we removed the statement about the number of species per clade from the Results and added it to the legend of Extended Data Fig. 6 (ancestral state reconstruction of biogeographic clades).

Originally: *Since the Lesser Antillean species are divided into two small and distantly related clades, each with relatively few species (nine and eleven, resp.; Fig. 1a), we*

excluded them from the following comparisons and instead focused on the three major clades.

Revised (lines 150-153): *Since the Lesser Antillean species are divided into two small and distantly related clades, each with relatively few species (resp.; Fig. 1a), we excluded them from the following comparisons and instead focused on the three major clades.*

Sentence added (lines 889-891): *Note that the Lesser Antillean clade consists of two clades plus a single species (*A. acutus*) that colonized the Lesser Antilles recently from the Greater Antilles.*

- In line 102, the wording is unclear regarding which hypothesis is H1 and which is H2. From the phrasing of the sentence, I would conclude that H1 has the pelvic and pectoral girdles forming a single evolutionary module. However, in Figure 2 H1 is clearly labelled as the hypothesis in which the pelvic and pectoral girdles are treated as separate. Could you edit this to keep it consistent?

Response: We thank the reviewer for picking this up. We have now clarified the specification of module structures.

Originally: *[...] they are expected to co-evolve more tightly than other parts of the locomotor skeleton and form one single evolutionary module (with the pelvis and pectoral forming a single or two separate modules; H1 and H2 in Fig. 2).*

Revised (lines 161-163): *[...] they are expected to co-evolve more tightly than other parts of the locomotor skeleton and form one single evolutionary module (with the pelvis and pectoral forming two separate modules (H1) or a single module (H2); Fig. 2).*

- I suggest rephrasing the sentence beginning on line 332 – maybe end it with “despite the genetic change being random” instead of the current phrasing.

Response: We have followed this advice and rephrased the sentence.

Originally: *[...] developmental interactions that make those traits vary together despite that genetic change is random⁶⁰⁻⁶² (reviewed in ref. 13).*

Revised (lines 299, 300): *[...] developmental interactions that make those traits vary together despite the genetic change being random⁶⁰⁻⁶² (reviewed in ref. 13).*

- In line 362: are there any species that you think may be candidates to have occupied those niches on the mainland? This isn't essential to your point, but it could be interesting.

Response: Unfortunately, we can only speculate at this point about the absence of some extreme *Anolis* morphologies on the mainland. Nevertheless, we have rephrased this sentence to include the possibility of other taxa than lizards to occupy these ecological niches.

Originally: *The apparent lack of these specialized morphologies on the mainland may reflect that those niches were already occupied by other taxa, or that grass-bush, crown-giant and twig ecomorphs are not viable on the mainland because of, for example, high predation pressure²⁹.*

Revised (lines 333-337): *The apparent lack of these specialized morphologies on the mainland may reflect that those niches were already occupied by other members of the*

rich continental lizard fauna, or possibly even other taxa. Alternatively, grass-bush, crown-giant and twig ecomorphs might not be viable on the mainland because of, for example, high predation pressure²⁹.

- In supplementary tables 3 & 4, it might be helpful to include some method to visually differentiate between the different values being reported. For example, minor shading of the Procustes variances would make it easier to interpret quickly.

Response: Thank you for this good suggestion. We have implemented the suggested change to help the reader in visually grasping these tables more quickly (also applied to Supplementary Tables 6-9 in addition to Supplementary Tables 3 and 4).

Reviewers' Comments:

Reviewer #1:

Remarks to the Author:

The authors have responded to my reviewer comments reasonably. I still think we differ in the use of "development" as an explanation for their results, although I agree it is a possibility. I also think they misread my comment. Development obviously includes pleiotropy and regulatory connections, but I think late stage functional integrations are fuzzier, and I don't think the term development applies to that case.

However, this issue does not need to be resolved here and is not critical to the conclusion.

I particularly appreciate their additional supplementary testing the effect of extreme morphologies on the results. That satisfied my question.

I have no further comments.

Reviewer #2:

Remarks to the Author:

I have re-read the manuscript by Feiner et al. I think that the minor modifications they have made have greatly improved this manuscript. I have no further concerns. I feel like this will make a strong and impressionable impact on the field of morphological evolution.

Below is the report of our manuscript revision. We show sentences taken from the manuscript in italic, and parts newly inserted into the manuscript as underlined. Page and line numbers refer to positions in the document with tracked changes.

Reviewer #1 (Remarks to the Author):

The authors have responded to my reviewer comments reasonably. I still think we differ in the use of “development” as an explanation for their results, although I agree it is a possibility. I also think they misread my comment. Development obviously includes pleiotropy and regulatory connections, but I think late stage functional integrations are fuzzier, and I don't think the term development applies to that case. However, this issue does not need to be resolved here and is not critical to the conclusion.

Response: We acknowledge the different definitions of the term ‘development’. To ensure that these are not misleading readers concerning how we interpret the results, we have rephrased several sentences in our Discussion. In particular, we now avoid the term developmental bias but instead spell out what exactly we mean.

Originally: *A role for developmental bias is supported by the fact that the clade that recolonized the mainland exhibits an equally strong covariation between limbs and girdles as seen in the Greater Antillean radiation.*

Revised (lines 285-287): *A role for persistent bias in the generation of phenotypic variation is supported by the fact that the clade that recolonized the mainland exhibits an equally strong covariation between limbs and girdles as seen in the Greater Antillean radiation.*

Furthermore, we have now replaced ‘development’ with ‘development and growth’ to avoid the impression that we are referring to embryological processes only.

Originally: *It is therefore plausible that the distinct evolutionary modularity of the Greater Antillean clade, relative to the Primary Mainland clade, in part reflects an ancient and persistent difference in how girdles and limbs develop together (‘developmental bias’¹³).*

Revised (lines 277-280): *It is therefore plausible that the distinct evolutionary modularity of the Greater Antillean group, relative to the Primary Mainland clade, in part reflects an ancient and persistent difference in how girdles and limbs develop and grow together.*

Originally: *Furthermore, if selection is able to modify developmental bias^{61,62,71}, species with distinct morphologies, like the Chamaeleonides group of the Greater Antillean clade, may stand out in terms of morphological variability.*

Revised (lines 318-321): *Furthermore, if selection is able to modify skeletal development and growth^{61,62,71}, species with distinct morphologies, like the Chamaeleonides group of the Greater Antillean Anolis, may stand out in terms of morphological variability.*

I particularly appreciate their additional supplementary testing the effect of extreme morphologies on the results. That satisfied my question.

Response: We thank the reviewer for the positive assessment of our additional analyses.

I have no further comments.

Response: We are grateful for the thorough review of our manuscript and thank the reviewer for her/his time.

Reviewer #2 (Remarks to the Author):

I have re-read the manuscript by Feiner et al. I think that the minor modifications they have made have greatly improved this manuscript. I have no further concerns. I feel like this will make a strong and impressionable impact on the field of morphological evolution.

Response: We are very glad about the positive evaluation and the encouraging words of the reviewer. We thank her/him for the constructive feedback.